# NoiGAN: Noise Aware Knowledge Graph Embedding with GAN

## Abstract

Knowledge graph has gained increasing attention in recent years for its successful applications of numerous tasks. Despite the rapid growth of knowledge construction, knowledge graphs still suffer from severe incompletion and inevitably involve various kinds of errors. Several attempts have been made to complete knowledge graph as well as to detect noise. However, none of them considers unifying these two tasks even though they are inter-dependent and can mutually boost the performance of each other. In this paper, we proposed to jointly combine these two tasks with a unified Generative Adversarial Networks (GAN) framework to learn noise-aware knowledge graph embedding. Extensive experiments have demonstrated that our approach is superior to existing state-of-the-art algorithms both in regard to knowledge graph completion and error detection.

## 1 Introduction

Knowledge graph, as a well-structured effective representation of knowledge, plays a pivotal role in many real-world applications such as web search (Graupmann et al., 2005), question answering(Hao et al., 2017; Yih et al., 2015), and personalized recommendation (Zhang et al., 2016). It is constructed by extracting information as the form of triple from unstructured text using information extraction systems. Each triple $(h, r, t)$ represents a relation $r$ between a head entity $h$ and a tail entity $t$. Recent years have witnessed extensive construction of knowledge graph, such as Freebase (Bollacker et al., 2008), DBPedia (Auer et al., 2007), and YAGO (Suchanek et al., 2007). However, these knowledge graphs suffer from severe sparsity as we can never collect all the information. Moreover, due to the huge volumes of web resources, the task to construct knowledge graph usually involves automatic mechanisms to avoid human supervision and thus inevitably introduces many kinds of errors, including ambiguous, conflicting and erroneous and redundant information.

To address these shortcomings, various methods for knowledge graph refinement have been proposed, whose goals can be arguably classified into two categories: (1) knowledge graph completion, the task to add missing knowledge to the knowledge graph, and (2) error detection, the task to identify incorrect triples in the knowledge graph. Knowledge graph embedding (KGE) currently hold the state-of-the-art in knowledge graph completion for their promising results (Bordes et al., 2013; Yang et al., 2014). Nonetheless, they highly rely on high quality training data and thus are lack of robustness to noise (Pujara et al., 2017). Error detection in knowledge graph is a challenging problem due to the difficulty of obtaining noisy data. Reasoning based methods are the most widely used methods for this task (Paulheim, 2017). Without the guidance of noisy data, they detect errors by performing reasoning over the knowledge graph to determine the correctness of a triple. A rich ontology information is required for such kind of methods and thus impede its application for real-world knowledge graphs.

Existing works consider knowledge graph embedding and error detection independently whereas these two tasks are inter-dependent and can greatly influence each other. On one hand, error detection model is extremely useful to prepare reliable data for knowledge graph embedding. On the other hand, high quality embedding learned by KGE model provides a basis for reasoning to identify noisy data. Inspired by the recent advances of generative adversarial deep models (Goodfellow et al., 2014), in this paper, we proposed to jointly combine these two tasks with a unified GAN framework, known as NoiGAN, to learn noise-aware knowledge graph embedding. In general, NoiGAN consists of two main components, a noise-aware KGE model to learn robuster representation of

knowledge and an adversarial learning framework for error detection. During the training, noise-aware KGE model takes the confidence score learned by GAN as guidance to eliminate the noisy data from the learning process whereas the GAN requires that KGE model continuously provides high quality embedding as well as credible positive examples to model the discriminator and the generator. Cooperation between the two components drives both to improve their capability. The main contributions of this paper are summarized as follows:

- We propose a unified generative adversarial framework NoiGAN, to learn noise-aware knowledge graph embedding. Under the framework, the KGE model and error detection model could benefit from each other: the error detection model prepares reliable data for KGE model to improve the quality of embedding it learns, while the KGE model provides a promising reasoning model for the error detection model to better distinguish noisy triples from the correct one.

- Our proposed framework can be easily generalized to various KGE models to enhance their ability in dealing with noisy knowledge graph.

- We experimentally demonstrate that our new algorithm is superior to existing state-of-the-art algorithms. The KGE model and GAN can alternately and iteratively boost performance in terms of both knowledge graph completion and noise detection.

## 2 RELATED WORK

### 2.1 KNOWLEDGE GRAPH COMPLETION

Embedding based methods currently hold the state-of-the-art in knowledge graph completion for their promising results (Wang et al., 2017b). They aim to capture the similarity of entities by embedding entities and relations into continuous low-dimensional vectors. Existing methods can be roughly divided into two categories: translational distance models and semantic matching models. Translational distance models measure the plausibility of a fact as the distance between two entities after a translation carried out by the relation. TransE (Bordes et al., 2013), TransH (Wang et al., 2014) and TransR (Lin et al., 2015) are the representative approaches in this category. Semantic matching models measure plausibility of facts by matching latent semantics of entities and relations embodied in their vector space representations. The typical models include RESCAL (Nickel et al., 2011), DistMult (Yang et al., 2014) and ComplEx (Trouillon et al., 2016). To optimize the KGE model,negative sampling is usually required to minimize the margin based ranking loss. A conventional method to construct negative samples is randomly sampling. However, negative samples generated through a random mode are often too easy to be discriminated from positive facts and thus make little contribute towards the training. Some recent works proposed to incorporate GAN for better negative sampling to improve the quality of embeddings (Wang et al., 2018; Cai & Wang, 2017). Nonetheless, none of the above methods has taken the potential noisy data into consideration, which leads to their sensitivity to unreliable data (Pujara et al., 2017). In this paper, we proposed a novel technique to enable current embedding models to cope with noisy data.

### 2.2 ERROR DETECTION IN KNOWLEDGE GRAPH

Due to lack of noisy data samples, error detection in knowledge graph is a challenging task. Existing methods can be either ontology based or anomaly detection based. Ontology based methods address this problem by exploring additional ontology information. A larger number of ontology reasoners are developed to utilize logic programming to derive uncovering contradictions among observed facts (Luther et al., 2009; Dentler et al., 2011; Ding et al., 2007). Rich ontology information is required for such kind of methods and thus impede its application to real-world knowledge graphs. Another kind of methods is anomaly detection based methods (Wienand & Paulheim, 2014; Fleischhacker et al., 2014). The main drawback of anomaly detection based methods are that they do not necessarily identify errors, but also natural outliers, which will compromise the objectivity of its results. More recently, a novel confidence-aware framework (Xie et al., 2018; Shan et al., 2018) proposed to incorporate triple confidence into KGE model to detect noises while learning knowledge representations simultaneously. However, it measures the triple confidence merely based on the how well the triple fits the model, which is easily affected by model bias.

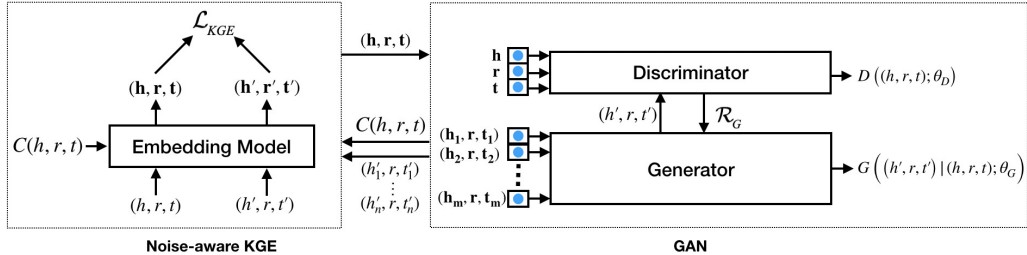

Figure 1: An Overview of the NoiGAN Framework.

# 3 PROPOSED METHOD:NOIGAN

In this section, we illustrate our proposed framework, NoiGAN, in details. The goal of our proposed framework is to learn noise-aware KG embedding with Generative Adversarial Networks (GAN). As shown in Figure 1, it consists of three components: (i) Noise aware KGE model, which incorporates a confidence score $C(h, r, t)$ into the KGE model to isolate the impact of noise over embedding vectors. Meanwhile, it provides high quality embedding to model discriminator and generator (Section 3.1); (ii) Triple generator $G((h', r, t')|(h, r, t); \theta_G)$, which learns to generate the most likely triples to be the noisy triples by corrupting the correct triple $(h, r, t)$. It aims at providing reliable noisy data for the discriminator as well as noise aware KGE model (Section 3.2.1); (iii) Discriminator $D((h, r, t); \theta_D)$, which tries to distinguish correct triples from noisy triples, and assigns a confidence score for each triple fact to describe the correctness of it. The discriminator is used to determine confidence score $C(h, r, t)$ in the noise aware KGE model. It also naturally provides guidance to the generator to produce higher quality noisy data (Section 3.2.2).

Before further discussing the detail of the algorithm, we first give a formal definition to knowledge graph. A knowledge graph, denoted by $\mathcal{G} = \{E, R, \mathcal{T}\}$, consists of a set of entities $E$, a set of relations $R$ and a set of of observed facts $\mathcal{T} = \{(h, r, t) \mid h, t \in E, \ r \in R\}$. A fact is represented by a triple $(h, r, t)$, where $h$, $t$ and $r$ denote head entity, tail entity, and relation, respectively.

## 3.1 NOISE AWARE KNOWLEDGE GRAPH EMBEDDING MODEL

To learn a more robust representation for noisy knowledge graph, we propose a noise-aware KGE model to eliminate the impact of noise over embedding vectors by incorporating confidence score into KGE model. It can be easily adapted to any KGE models. For example, we can follow TransE to represent entities and relations. Given a fact $(h, r, t)$, TransE aims to learn corresponding low dimensional vectors $\mathbf{h}$, $\mathbf{r}$ and $\mathbf{t}$ in the same space for $h$, $r$ and $t$, so that the distance between $\mathbf{h} + \mathbf{r}$ and $\mathbf{t}$ is minimized. The scoring function of TransE is then defined as $f_r(h, t) = \|\mathbf{h} + \mathbf{r} - \mathbf{t}\|_1^2$. To optimize the KGE model, we use a loss function similar to the negative sampling loss in (Mikolov et al., 2013) according to (Sun et al., 2019).

$$\sum_{(h,r,t)\in\mathcal{T}} [(-\log\sigma(\gamma - f_r(h,t)) + \sum_{(h',r,t')\in\mathcal{N}(h,r,t)} \frac{1}{|\mathcal{N}(h,r,t)|}\log\sigma(\gamma - f_r(h',t')))]$$

where $\gamma$ is the margin, $\sigma$ is the sigmoid function, $\mathcal{T}$ represents the observed facts from the knowledge graph and $(h', r, t')$ is the negative sample for triple $(h, r, t)$. To reduce the effects of randomness, we sample multiple negative triples for each observed fact. We denote the negative triples set of a triple $(h, r, t)$ as $\mathcal{N}(h, r, t)$. Negative triples set is constructed by replacing the head or tail entity of the observed triples with an entity sampled randomly from entity set $E$:

$$\mathcal{N}(h,r,t) = \{(h',r,t)|h' \in E\} \cup \{(h,r,t')|t' \in E\}, (h,r,t) \in \mathcal{T}$$

A vast majority of existing embedding models assume that all triple facts hold true to knowledge graph, which is inappropriate. In fact, knowledge graph contains many kinds of errors (e.g., ambiguous, conflicting, erroneous, redundant) due to the automatic construction process. To isolate the

impact of noise over embedding vectors, following (Shan et al., 2018), we adopted the concept **confidence** to describe whether a triple fact is noisy or not. In (Shan et al., 2018), confidence of a triple is measured by how well the triple fits to the KGE model. A central issue with such measurement is that it is easily affected by model bias and thus lead to unreliability. Unlike (Shan et al., 2018), we learned the confidence scores using a discriminator, which can be more impartial and promising. We will discuss how to obtain such discriminator in Section 3.2.2. With the introduction of confidence, we can eliminate the noisy data from the learning process of KGE model.

$$\mathcal{L}_{KGE} = \sum_{(h,r,t)\in\mathcal{T}} C(h,r,t).[(-\log\sigma(\gamma-f_r(h,t)) + \sum_{(h',r,t')\in\mathcal{N}(h,r,t)} \frac{1}{|\mathcal{N}(h,r,t)|} \log\sigma(\gamma-f_r(h',t')))]$$

where $C(h,r,t)$ is the confidence score assigned to each positive triple $(h,r,t)$. To be specific, $C(h,r,t)$ can be both a binary variable or a soft value from the interval $[0,1]$. We denote the previous one as the hard version and the later one as the soft version.

## 3.2 GAN FOR NOISE IDENTIFICATION

After introducing our noise aware KGE model, we now show how to identify the error in knowledge graph to enhance embedding quality. Typically label information about noisy triples is costly and labor intensive to obtain, which is the central challenge to the task of error detection. Without the guidance of noisy data, previous works propose to leverage ontology information to exploit reasoning for inconsistency detection. However, these ontology information can also be difficult to gain. To address this problem, inspired by recent advance of GAN, we propose a novel end-to-end adversarial learning framework to detect noise in knowledge graph. The proposed GAN cooperates with the noise aware KGE model in an interactive way. GAN utilizes the embedding learned by the noise aware KGE model to distinguish true triples and noisy triples. Meanwhile the noise-aware KGE model requires GAN to learn a confidence score so as to eliminate the impact of noise over embedding vectors.

Our adversarial learning framework consists of two components, a generator and a discriminator. Unlike traditional GAN whose ultimate goal is to train a good generator, we aim to produce a good discriminator to distinguish noisy triples from true triples. To address noisy data deficiency issue, the generator and the discriminator act as two players in a minimax game: the generator learns to continually generate "more confusing" triples and thus provides better quality noisy data for the discriminator, whereas the discriminator is trained to draw a clear distinction between the true triples and the noisy triples generated by its opponent generator. Formally, it can be formulated as follows:

$$\max_D \min_G V(G,D) = \sum_{(h,r,t)\in\mathcal{T}} \mathbb{E}_{(h,r,t)\sim\mathcal{T}}[\log D((h,r,t);\theta_D)]$$
$$+ \mathbb{E}_{(h',r,t')\sim G(\cdot|(h,r,t);\theta_G)}[\log(1 - D((h',r,t');\theta_D))]$$

where $\mathcal{T}$ represents the true triples set. To get rid of the noise in knowledge graph, we construct $\mathcal{T}$ with top 10% triples which fit the KGE the best. We discuss the detailed implementation of the generator and discriminator as follows.

### 3.2.1 TRIPLE GENERATOR

To identify the error in knowledge graph, first we have to achieve reliable examples of noisy data. Since such data is costly and labor intensive to obtain, we instead introduce a generator to produce them. The main goal of the generator is to generator high-quality fake triples that can fool discriminators, which aims at bringing a better classifier to identify error in knowledge graph. In addition, to simultaneously make KGE model aware of the existing noisy data, we also explicitly incorporate the noisy triples generated by the generator as negative samples to train the noise aware KGE model.

Note that, the generator takes embedding vectors learned by the noise-aware KGE model as input. Given a true triples $(h,r,t)$, the generator aims to select the most likely triples to be the noisy triples $(h',r,t')$ from its negative samples candidate set $\mathcal{N}(h,r,t)$. To achieve this goal, a two-layer fully-connected neural network is introduced to model the probability distribution over the candidate

negative samples $\mathcal{N}(h, r, t)$. In particular, this MLP uses ReLU as the activation function for the first layer and adds Softmax to the second layer. It takes the concatenation of the embedding vectors $\mathbf{h}'$, $\mathbf{r}$ and $\mathbf{t}'$ of triple $(h', r, t')$ as input and output the probability whether the triple $(h', r, t')$ is the most likely noisy triple. As the output of the generator is a discrete triple, training for the generator is a little tricky due to data indifferentiability issue. A common solution is to use policy gradient based reinforcement learning instead (Wang et al., 2017a). In particular, for a generated triple, the reward from discriminator is defined as $f_D(h', r, t')$, which is exactly the probability of the triple $(h', r, t')$ to be true. Thus, in order to fool the discriminator, the generator is trained to maximize the expected reward as follows:

$$\mathcal{R}_G = \sum_{h,r,t} \mathbb{E}_{(h',r,t') \sim G(\cdot|(h,r,t);\theta_G)}[\log f_D(h', r, t')] \tag{1}$$

At the very beginning, because of lack of guidance, the generator generates "noise" by random sampling. However, such "noise" is too easy for discriminator to distinguish and result in a low reward. By learning to maximize the reward, the generator continually learns better probability distribution of the candidate negative samples and keep generating "more confusing" triples to improve the capacity of the discriminator. Moreover, this generator can also be used to produce high quality negative samples for the noise aware KGE model.

### 3.2.2 DISCRIMINATOR

Discriminator aims to distinguish the true triples from the noisy triples. Both positive and negative examples are required to train the discriminator. Generator learns to provide reliable negative examples for the discriminator while observed triples in knowledge graph can be regarded as positive examples. However, as knowledge graph is noisy, considering all observed triples as positive samples will inevitably introduce noise. We therefore utilize an interesting observation, when noisy data exist, deep learning models tend to memorize these noise in the end, which leads to the poor generalization performance (Han et al., 2018). Specifically, we take top $10\%$ triples which fit the KGE the best (with lowest $f_r(h, t)$) as positive training examples. Considering discriminator is essentially a binary classifier, the objective function of the discriminator can be formulated as minimizing the following cross entropy loss:

$$\mathcal{L}_D = - \sum_{(h,r,t) \in \mathcal{T}} \log(f_D(h, r, t)) - \sum_{(h',r,t') \in G(\cdot|(h,r,t);\theta_G)} \log(1 - f_D(h', r, t')) \tag{2}$$

where $f_D(h, r, t) = \sigma(\text{MLP}(h, r, t))$ where MLP is a two-layer neural network with ReLU as activation function and $\sigma(x) = 1/(1 + \exp(-x))$ is the standard sigmoid function. If we use TransE as our KGE model, the MLP takes the vector $\mathbf{h} + \mathbf{r} - \mathbf{t}$ as input and output the probability of the triple $(h, r, t)$ being true. $f_D(h, r, t)$ will later be used to define $C(h, r, t)$ for each positive triple $(h, r, t)$ in knowledge graph. To be specific, $C(h, r, t)$ can be either a binary value or a soft value. When it is a binary variable, it represents the classification result regard to whether a triple is a true triple. When it takes a soft value, it indicates the probability of the triple $(h, r, t)$ being true. We denote the previous one as a hard version of our proposed model and the later one as the soft version.

---

**Algorithm 1** NoiGAN Framework

---

**Require:** Observed triples in knowledge bases $\mathcal{T}$
 1: Initialize $C(h, r, t)$ as 1 for all triples.
 2: Train noise aware KGE model with random negative sampling until convergence.
 3: **for** $n = 1 : N$ **do**
 4:     Take top $10\%$ triples which fit the noise aware KGE the best as positive examples.
 5:     Train GAN with the selected positive examples until convergence.
 6:     Update $C(h, r, t)$ according to discriminator of GAN.
 7:     Train noise aware KGE model with negative samples generated by the generator until convergence.
 8: **end for**

---

| Dataset | #Relations | #Entities | #Train | #Valid | #Test |
|---------|-----------|-----------|--------|--------|-------|
| FB15K | 1,345 | 14,951 | 483,142 | 50,000 | 59,071 |
| WN18 | 18 | 40,943 | 141,442 | 5,000 | 5,000 |
| FB15K-237 | 237 | 14,541 | 272,115 | 17,535 | 20,466 |
| WN18RR | 11 | 40,943 | 86,835 | 3,034 | 3,134 |
| YAGO3-10 | 37 | 123,182 | 1,079,040 | 5,000 | 5,000 |

Table 1: Data Statistics

## 3.3 JOINT TRAINING

NoiGAN is trained in an iterative way, through which each component of the model can alternately and iteratively boost performance of each other. The training process is summarized in Algorithm 1. We train each component of NoiGAN alternately by fixing the parameters of the other components. We start by training the noise aware KGE model. We initiate the confidence score $C(h, r, t)$ as 1 for all triple facts and pretrain the noise aware KGE model with random negative sampling to gain reliable embedding vector. With these embedding vectors, we start the training of GAN model from the generator. The generator learns to understand the probability distribution of the candidate negative samples so as to select the most likely noisy data. In particular, the discriminator gives it a reward as guidance. Once the generator finishes training, it can be used to prepare noisy data for the discriminator. In addition, we take top $10\%$ triples facts which fit the noise aware KGE the best as credible triple facts. With noisy data as negative examples and triple facts as positive examples, discriminator learns a classifier to distinguish the positive triples from the negative triples. After we obtain the well trained GAN model, given the confidence score assigned by the discriminator and the negative samples generated by the generator, we retrain the noise aware KGE model and update the embedding vectors. The process repeats until convergence. The embeddings learned by the noise aware KGE model is regarded as our final representation for entities and relations.

## 4 EXPERIMENTS

### 4.1 EXPERIMENTAL SETTINGS

**Datasets.** We evaluate our NoiGAN on five benchmark datasets, including FB15K, WN18 ,FB15K-237, WN18RR and YAGO3-10. These benchmark datasets benefit from human curation that results in highly reliable facts. To simulate the real-world knowledge graphs extracted automatically from unstructured text data, we modified these benchmark datasets to include noisy triples. Since all kinds of noise might be contained while we construct knowledge graphs, our approach to introducing noise is to substitute the true head entity or tail entity with any randomly selected entity. Following this approach, we construct five KGs based on each benchmark dataset with noisy triples to be different ratio of (e.g., 40%, 70% and 100%) of true triples. All noisy datasets share the same entities, relations, validation and test sets with the original benchmark dataset, with all generated noisy triples fused into the original training set. The statistics of these knowledge graphs are summarized in Table 1.

**Baselines.** NoiGAN is compared with the following state-of-the-art algorithm, including (1) KGE models (e.g., TransE (Bordes et al., 2013),DistMult (Yang et al., 2014) and RotatE (Sun et al., 2019)), (2) robust KGE models ( e.g., attention based method (Nathani et al., 2019)), (3) noise aware KGE models (e.g., CKRL (Xie et al., 2018)) and (4) KGE models with GAN (e.g., KBGAN (Cai & Wang, 2017)). In particular, there are three kinds of triple confidences defined in CKRL (Xie et al., 2018). In our paper, we take CKRL with local triple confidence, called CKRL (LT), as baseline. To fairly compare different methods, the same loss function and negative sampling strategies are employed for all models.

**Experimental Setup of NoiGAN.** We evaluate two versions of our NoiGAN model as mentioned in Section 3.2.2. The soft version is denoted as NoiGAN (soft) while the hard version is denoted as NoiGAN (hard). To show that our NoiGAN can be easily generalized to various KGE models, TransE (Bordes et al., 2013) and RotatE (Sun et al., 2019) are implemented as score function for NoiGAN. Adam (Kingma & Ba, 2014) is adopted as the optimizer. We set the parameters for all

| | Dataset | FB15K | | YAGO3-10 | | WN18 | |
|---|---|---|---|---|---|---|---|
| | Metric | AUC | Spe | AUC | Spe | AUC | Spe |
| 10% | CKRL (LT) | .548 | .006 | .455 | .018 | .704 | .000 |
| | NoiGAN-TransE (soft) | **.949** | .791 | .929 | .625 | .794 | .452 |
| | NoiGAN-TransE (hard) | .946 | **.821** | **.942** | **.730** | **.836** | **.565** |
| 20% | CKRL (LT) | .687 | .008 | .521 | .000 | .710 | .000 |
| | NoiGAN-TransE (soft) | .933 | .733 | .914 | .619 | .756 | .336 |
| | NoiGAN-TransE (hard) | **.954** | **.823** | **.934** | **.711** | **.799** | **.524** |
| 40% | CKRL (LT) | .725 | .018 | .577 | **1.000** | .725 | .000 |
| | NoiGAN-TransE (soft) | .931 | .699 | .894 | .552 | .682 | .345 |
| | NoiGAN-TransE (hard) | **.950** | **.780** | **.929** | .711 | **.741** | **.464** |

Table 2: Evaluation Results on Knowledge Graph Error Detection.

methods by a grid search strategy. The range of different parameters is set as follows: embedding dimension $k \in \{250, 500, 1000\}$, batch size $b \in \{256, 512, 1024\}$, and fixed margin $\gamma \in \{9, 12, 24\}$. Afterwards, we compare the best results of different methods. Both the entity embeddings and the relation embeddings are uniformly initialized and no regularization is imposed on them. As mention in Section 3.2, we implement both discriminator and the generator as simple two-layer fully connected neural networks. The size of hidden states for each of the two networks is set to 10.

## 4.2 KNOWLEDGE GRAPH NOISE DETECTION

To verify the capability of NoiGAN in distinguishing noises in knowledge graphs, we evaluate NoiGAN in terms of classification performance. To be specific, we classify the training data by determining whether a triple is being true. For NoiGAN (hard), we can directly utilize the discriminator to classify noise. Whereas, for NoiGAN (soft), the discriminator only assigns a soft score between 0 to 1 to each triple, indicating the probability of a triple being true. We thus classify the triples by regarding the triple who have $C(h, r, t) > 0.5$ as the true triple and the remaining as noise. The experiments are conducted on three benchmark datasets, including FB15K, WN18 and YAGO3-10 with noisy triples to be different ratio of 10%, 20% and 40% of true triples. Although FB15K and WN18RR suffer from test triple leakage in the training set (Dettmers et al., 2018), we report the results over training data, which won't be affected by the issue. As we make similar observations on NoiGAN-TransE and NoiGAN-RotatE, we only report the experimental results w.r.t. NoiGAN-TransE to save space.

**Evaluation Metric.** Two classification evaluation metrics are adopted, including (1) AUC; (2) A special measurement defined as the proportion of actual noises that are correctly identified. It can be calculated as $\frac{TN}{TN+FP}$, where TN represents true negative while FP represents false positive.

**Results.** The results w.r.t. NoiGAN-TransE can be found in Table 2. We compared against CKRL (LT) as it is the only baseline method which is able to detect noise in the training dataset. We can observe that: (1) Our model consistently achieves the best performances over all datasets in all cases, which demonstrates the capability of our models in detecting noises. (2) NoiGAN-TransE (hard) has significant improvements in noise detection compared to NoiGAN-TransE (soft). Considering that NoiGAN-TransE (soft) merely assigns low confidence scores to noisy triples while NoiGAN-TransE (hard) completely eliminates these data, this result further validates that our method can classify noisy data accurately. (3) The baselines tend to classify all triples into the same class, which shows their inability in detecting noise.

## 4.3 KNOWLEDGE GRAPH COMPLETION

In addition to its promising results on noise detection, our approach is also superior to existing state-of-the-art algorithms in terms of the quality of learned embedding. We conduct experiments on three benchmark datasets, including FB15K-237, WN18RR and YAGO3-10 with noisy triples to be different ratio of 40%, 70% and 100% of true triples.

| | | FB15K-237 | | | | YAGO3-10 | | | | WN18RR | | | |
| --- | --- | --- | --- | --- | --- | --- | --- | --- | --- | --- | --- | --- | --- |
| | | MRR | H@1 | H@3 | H@10 | MRR | H@1 | H@3 | H@10 | MRR | H@1 | H@3 | H@10 |
| 0% | DistMult | .218 | .136 | .236 | .388 | .292 | .203 | .325 | .470 | .424 | .382 | .437 | .507 |
| | KBGAN | .266 | .186 | .290 | .427 | .071 | .041 | .070 | .124 | .215 | .036 | .356 | .507 |
| | Attention | **.436** | **.360** | **.464** | **.589** | - | - | - | - | .443 | .363 | **.489** | **.583** |
| | TransE | .330 | .229 | .371 | .531 | .410 | .299 | .469 | **.620** | .229 | .022 | .403 | .526 |
| | CKRL (LT) | .330 | .229 | .372 | .529 | .377 | .264 | .438 | .585 | .231 | .027 | .404 | .530 |
| | NoiGAN-TransE (soft) | .318 | .217 | .358 | .518 | .370 | .267 | .420 | .567 | .231 | .026 | .407 | .524 |
| | NoiGAN-TransE (hard) | .308 | .209 | .349 | .502 | .345 | .248 | .393 | .531 | .223 | .031 | .381 | .503 |
| | RotatE | .326 | .228 | .363 | .521 | .380 | .269 | .437 | .594 | **.472** | **.429** | .488 | .557 |
| | NoiGAN-RotatE (soft) | .326 | .229 | .363 | .520 | **.430** | **.330** | **.487** | **.620** | .466 | .420 | .485 | .554 |
| | NoiGAN-RotatE (hard) | .325 | .228 | .363 | .518 | .413 | .312 | .471 | .602 | .465 | .420 | .483 | .551 |
| 40% | DistMult | .207 | .127 | .229 | .369 | .089 | .038 | .095 | .188 | .364 | .310 | .400 | .449 |
| | KBGAN | .177 | .079 | .214 | .363 | .059 | .033 | .061 | .102 | .178 | .010 | .312 | .447 |
| | Attention | **.326** | **.249** | **.352** | .480 | - | - | - | - | .230 | .035 | .357 | **.586** |
| | TransE | .182 | .026 | .271 | .463 | .150 | .001 | .231 | .429 | .185 | .008 | .331 | .455 |
| | CKRL (LT) | .190 | .035 | .276 | .471 | .122 | .007 | .175 | .344 | .186 | .010 | .327 | .462 |
| | NoiGAN-TransE (soft) | .212 | .064 | .294 | .482 | .109 | .004 | .156 | .306 | .188 | .005 | .337 | .469 |
| | NoiGAN-TransE (hard) | .227 | .095 | .296 | .473 | **.321** | **.204** | **.383** | **.538** | .168 | .008 | .300 | .407 |
| | RotatE | .284 | .189 | .322 | .469 | .098 | .034 | .116 | .228 | .399 | **.345** | .438 | .481 |
| | NoiGAN-RotatE (soft) | .294 | .197 | .331 | .488 | .220 | .120 | .266 | .420 | .403 | .339 | **.447** | .506 |
| | NoiGAN-RotatE (hard) | .296 | .198 | .336 | **.489** | .223 | .121 | .269 | .423 | **.403** | .339 | .445 | .507 |
| 70% | DistMult | .196 | .122 | .215 | .342 | .074 | .030 | .077 | .160 | .328 | .266 | .378 | .420 |
| | KBGAN | .123 | .040 | .145 | .288 | .026 | .016 | .027 | .040 | .161 | .007 | .276 | .429 |
| | Attention | .255 | .174 | .280 | .414 | - | - | - | - | .192 | .012 | .323 | .453 |
| | TransE | .167 | .019 | .244 | .440 | .085 | .005 | .106 | .242 | .169 | .005 | .300 | .439 |
| | CKRL (LT) | .171 | .021 | .253 | .449 | .090 | .004 | .119 | .262 | .169 | .007 | .299 | .441 |
| | NoiGAN-TransE (soft) | .189 | .037 | .274 | .464 | .124 | .003 | .183 | .358 | .166 | .002 | .289 | .442 |
| | NoiGAN-TransE (hard) | .199 | .059 | .275 | .457 | **.210** | **.091** | **.268** | **.443** | .152 | .004 | .267 | .392 |
| | RotatE | .254 | .154 | .299 | .444 | .037 | .001 | .044 | .103 | .357 | .292 | .411 | .456 |
| | NoiGAN-RotatE (soft) | .279 | .179 | **.321** | .475 | .121 | .034 | .154 | .296 | .366 | .295 | .423 | .473 |
| | NoiGAN-RotatE (hard) | **.282** | **.184** | **.321** | **.478** | .128 | .040 | .160 | .310 | **.368** | .298 | **.424** | **.474** |
| 100% | DistMult | .196 | .122 | .215 | .342 | .064 | .021 | .067 | .147 | .312 | .241 | .370 | .422 |
| | KBGAN | .091 | .030 | .099 | .213 | .009 | .002 | .008 | .021 | .138 | .010 | .218 | .392 |
| | Attention | .192 | .112 | .221 | .339 | - | - | - | - | .131 | .003 | .214 | .335 |
| | TransE | .157 | .016 | .227 | .422 | .065 | .003 | .076 | .191 | .153 | .002 | .267 | .422 |
| | CKRL (LT) | .161 | .018 | .233 | .430 | .070 | .002 | .083 | .211 | .151 | .005 | .250 | .419 |
| | NoiGAN-TransE (soft) | .173 | .020 | .258 | .454 | .100 | .001 | .145 | .286 | .147 | .000 | .247 | .415 |
| | NoiGAN-TransE (hard) | .185 | .044 | .262 | .446 | **.144** | **.016** | **.213** | **.384** | .129 | .004 | .223 | .339 |
| | RotatE | .234 | .137 | .276 | .418 | .019 | .001 | .020 | .051 | .315 | .239 | .383 | .429 |
| | NoiGAN-RotatE (soft) | **.283** | .201 | **.312** | .447 | .022 | .002 | .024 | .061 | **.327** | .240 | **.401** | .453 |
| | NoiGAN-RotatE (hard) | .271 | **.175** | .309 | **.461** | .021 | .002 | .022 | .058 | **.327** | **.244** | .399 | **.452** |

Table 3: Evaluation Results on Knowledge Graph Completion

| | Triple | Distance (TransE) | Distance (NoiGAN) | Score (NoiGAN) |
|---|---|---|---|---|
| Positive | (McDonald's (business), /dining/restaurant/cuisine, Philippine cuisine (national cuisine)) | 19.94 | 16.60 | 0.8664 |
| Negative | (McDonald's (business), /dining/restaurant/cuisine, Joe Walsh (human)) | 20.51 | 32.67 | 0.0171 |
| Positive | (voice actor (filmmaking occupation), /people/profession/people_with_this_profession, Michael Dobson (human)) | 21.58 | 20.28 | 0.8409 |
| Negative | (head teacher (profession), /people/profession/people_with_this_profession, Michael Dobson (human)) | 22.60 | 37.57 | 0.0058 |

Table 4: Examples of Noisy Triples and Positive Triples in FB15K Dataset.

**Evaluation Metric.** We mask the head or tail entity of each test triple, and require each method to predict the masked entity. During evaluation, we use the filtered setting (Bordes et al., 2013). The Hit@K (H@K) and Mean Reciprocal Rank (MRR) are adopted as the evaluation metrics.

**Results.** Results of reasoning are shown in Table 3. In particular, as attention based method exceed the memory capacity of our machines on YAGO3-10 dataset, only the results on the FB15K-237 and WN18RR datasets are reported. Note that TransE, CKRL (LT) and NoiGAN-TransE share the same score function TransE, they are comparable with each other. Similarly, RotatE and NoiGAN-RotatE share the same score function RotatE and thus they are comparable with each other. We make the following observations: (1) NoiGAN consistently outperforms the baseline methods which share the same score function with it on noisy dataset sets. The performance gain is significant especially on datasets with 100% noise. (2) Either NoiGAN-TransE or NoiGAN-RotatE will achieve the best performance on almost all noisy datasets. In particular, hard version of NoiGAN performs better then soft version in most cases. (3) Even though attention based method claims that they can ensure robust performance, the results show that our NoiGAN significantly outperforms it in terms of robustness. (4) If we do not introduce noise, the performance of NoiGAN models is almost the same as their variants. In addition, the improvement introduced by the NoiGAN becomes more significant as the noise rate in KGs rises. This further proves the robustness of the NoiGAN.

### 4.4 CASE STUDY

To demonstrate the power of discriminator in distinguishing noisy triples, we present some triples and their confidence score in Table 4. We conduct experiments over FB15K with 40% noise using the NoiGAN-TransE (hard) model. All hyperparameters are set as described in Section 4.1. To be clear, distance of a triple is calculated as $\|\mathbf{h} + \mathbf{r} - \mathbf{t}\|_1$ while the scores of the triples are learned by the discriminator. As shown in 4, we can see that TransE performs poorly at detecting some "triky" errors, such as logic error (e.g., *(head teacher, /people/profession/people_with_this_profession, Michael Dobson)*) and grammar error (e.g. *(McDonalds, /dining/restaurant/cuisine, Joe Walsh)*,). To our surprise, NoiGAN (hard) has the ability to detect both.

### 5 CONCLUSION

In this paper, we propose a novel framework NoiGAN, to jointly combine the tasks of knowledge graph completion and error detection for noise aware knowledge graph embedding learning. It consists of two main components, a noise-aware KGE model for knowledge graph completion and an adversarial learning framework for error detection. Under the framework, the noise-aware KGE model and the adversarial learning framework can alternately and iteratively boost performance of each other. Extensive experiments show the superiority of our proposed NoiGAN both in regard to knowledge graph completion and error detection.

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
