# OpenReview forum: "NoiGAN: NOISE AWARE KNOWLEDGE GRAPH EMBEDDING WITH GAN"
_ICLR.cc/2020/Conference — Reject_

### Official Review · AnonReviewer1 · 2019-10-16
**Official Blind Review #1**

**Rating:** 3

**Review:**

This paper proposes to provide a novel noise-aware knowledge graph embedding (NoiGAN) by combining KG completion and noise detection through the GANs framework. More specifically, NoiGAN repeatedly utilizes a GAN model to 1) approximate the confidence score for facts identifying reliable data (discriminator) and 2) generate more challenging negative samples (generator). Then, it uses this confidence score and negative samples to train a more accurate link prediction model. The authors validate the proposed model through several experiments.

This paper reads well and the results appear sound. I personally find the idea of incorporating confidence score into a link prediction model to achieve a more accurate model very interesting. Furthermore, the provided experiments support their intuition and arguments outperforming considered baselines.

As for the drawbacks, I find the baselines considered in this work outdated missing many SOTA and related works in link prediction and noise detection [1,2, 3, 4, 5]. Further, I believe this work needs more experimental results and an ablation study capturing different aspects of the presented method. My concerns are as follows:

•	Considering the existing reverse relation issue in FB15K and WN18, I suggest conducting the experiments on the FB15K-237 and WN18RR from [6] instead.
•	I suggest considering more recent link prediction models as baselines.
•	I am wondering if the only difference between NoiGAN and KBGAN [7] is incorporating the confidence score in the link prediction loss?
•	Considering the fact that NoiGAN repeatedly retrains GAN and link prediction model, I suggest providing a comparison of computational complexity.
•	I am wondering if NoiGAN can only work with pre-knowledge of noisy triples in KG? If not, why didn’t you report NoiGAN performance with 0% noise in Table 3?
•	I find utilizing few examples to evaluate the power of discriminator in distinguishing noisy triples (Table 4) not satisfactory at all. I suggest experimenting with more data and providing the per-relation breakdown performance of the discriminator.

On overall, although I find the proposed model quite novel and interesting, the paper needs more experimental results to validate the idea.

[1] Pinter, Yuval, and Jacob Eisenstein. "Predicting Semantic Relations using Global Graph Properties".
[2] Nathani, Deepak, et al. "Learning Attention-based Embeddings for Relation Prediction in Knowledge Graphs".
[3] Balažević, Ivana, Carl Allen, and Timothy M. Hospedales. "TuckER: Tensor Factorization for Knowledge Graph Completion".
[4] Sun, Zhiqing, et al. "Rotate: Knowledge graph embedding by relational rotation in complex space".
[5] Pezeshkpour, Pouya, Yifan Tian, and Sameer Singh. "Investigating Robustness and Interpretability of Link Prediction via Adversarial Modifications".
[6] Dettmers, Tim, et al. "Convolutional 2d knowledge graph embeddings.", AAAI-18.
[7] Liwei Cai and William Yang Wang. “Kbgan: Adversarial learning for knowledge graph embeddings”.


**Experience Assessment:**

I have published in this field for several years.

**Review Assessment: Checking Correctness Of Derivations And Theory:**

I assessed the sensibility of the derivations and theory.

**Review Assessment: Checking Correctness Of Experiments:**

I carefully checked the experiments.

**Review Assessment: Thoroughness In Paper Reading:**

I read the paper thoroughly.

---

> ### Author Response · Authors · 2019-11-15
> **Responds to Review #1**
>
> We thank the reviewer for the constructive reviews. We addressed the questions and concerns of the reviewer accordingly in the following.
>
> (1) Thanks to the reviewer for pointing out the problem of data leakage in FB15K and WN18. We have conducted the experiments on the FB15K-237 and WN18RR instead. Please find the result in Table 3 in our latest version of the paper.
>
> (2) Thanks to the reviewer for pointing the issue of insufficient baselines. We have added more baseline methods, including (1) KGE models (e.g., DistMult [5] and RotatE [4]), (2) robust KGE models (e.g., attention based method [6]) and (3) KGE models with GAN (e.g., KBGAN [3]). In addition, to show that our NoiGAN can be easily generalized to various KGE models, RotatE is also added as score function for NoiGAN. Please find the result in Table 3 in our latest version of the paper. The results show that both NoiGAN-TransE and NoiGAN-RotatE consistently and significantly outperform all the baseline methods in terms of robustness.
>
> (3) The goal of NoiGAN and KBGAN is totally different. KBGAN incorporates GAN for better negative sampling to improve the quality of embeddings. The discriminator of the GAN is their final KGE model.  However, in our case, NoiGAN utilizes GAN to determine whether a triple is noisy and it is independent with our KGE model. Different from KBGAN, the discriminator in our GAN is a binary classifier, which is used to learn confidence score for each triple to enable NoiGAN to cope with noisy training data. To further show the difference between NoiGAN and KBGAN, we have added KBGAN as our baseline and report the results in Table 3 in our latest version of the paper. The results indicate that KBGAN cannot cope with noisy training data.
>
> (4) To analyze the efficiency of NoiGAN, we compare the total training time until converge against the baseline methods on FB15K-237 with 100% noise as follows.
>
> Methods                                    The whole training time until converge (min)
> TransE [8]                                                                     60
> CKRL [7]                                                                      150
> DistMult [5]                                                                  40
> RotatE [4]                                                                     60
> KBGAN [3]                                                                    30
> attention based method [6]                                   3600
> NoiGAN-TransE                                                           60
>
> We can observe that our NoiGAN does not cost much time compared to other baseline methods.
>
> (5) Thanks to the reviewer for pointing the issue of not reporting NoiGAN performance with 0% noise. We have added these experiments as shown in Table 3 in our latest version of the paper. We can observe that for NoiGAN-RotatE, it has almost the same performance as its variant RotatE on FB15K-237 and WN18RR. It performs even better than RotatE on YAGO3-10. The major reason could be that YAGO3-10 contains more noise than FB15K-237 and WN18RR. Our NoiGAN-RotatE shows its superiority in this situation.
>
> [1]  “GraphGAN: Graph Representation Learning with Generative Adversarial Nets.” AAAI'18.
> [2]  “Irgan: A minimax game for unifying generative and discriminative information retrieval models.” SIGIR'17.
> [3]  “Kbgan: Adversarial learning for knowledge graph embeddings.” NAACL’18.
> [4]  “RotatE: Knowledge Graph Embedding by Relational Rotation in Complex Space.” ICLR'19.
> [5]  “Embedding Entities and Relations for Learning and Inference in Knowledge Bases.” ICLR'15.
> [6]  “Learning Attention-based Embeddings for Relation Prediction in Knowledge Graphs.” ACL’19
> [7]  “Does william shakespeare really write hamlet?  knowledge representation learning with confidence.” AAAI’18.
> [8] “Translating embeddings for modeling multi-relational data.” NeurIPS’13

---

### Official Review · AnonReviewer2 · 2019-10-25
**Official Blind Review #2**

**Rating:** 1

**Review:**

This paper proposes a GAN-oriented framework for training robust-to-noise neural link predictors. My main concern is that CKRL is the only baseline -- ignoring years of prior works in this space (see e.g. [1, 2]).
Furthermore, [2] shows that two of the three datasets used by the authors suffer from test triple leakage in the training set.

Finally, the considered datasets do not really test for the presence of noise - authors may want to check out e.g. https://arxiv.org/abs/1812.00279 (there are several works in this space, all of which were systematically ignored by this paper).

Finally, authors claim neural link predictors were never used for denoising, but actually [3] use them to learn a prior distribution over triples in a probabilistic DB setting.


[1] https://arxiv.org/abs/1806.07297
[2] https://arxiv.org/abs/1707.01476
[3] https://ai.google/research/pubs/pub45634

**Experience Assessment:**

I have published in this field for several years.

**Review Assessment: Checking Correctness Of Derivations And Theory:**

I carefully checked the derivations and theory.

**Review Assessment: Checking Correctness Of Experiments:**

I assessed the sensibility of the experiments.

**Review Assessment: Thoroughness In Paper Reading:**

I read the paper at least twice and used my best judgement in assessing the paper.

---

> ### Author Response · Authors · 2019-11-15
> **Responds to Review #2**
>
> We thank the reviewer for the constructive reviews. We addressed the questions and concerns of the reviewer accordingly in the following.
>
> (1) Thanks to the reviewer for pointing the issue of insufficient baselines. We have added more baseline methods, including (1) KGE models (e.g., DistMult [5] and RotatE [4]), (2) robust KGE models (e.g., attention based method [6]) and (3) KGE models with GAN (e.g., KBGAN [3]). In addition, to show that our NoiGAN can be easily generalized to various KGE models, RotatE is also added as score function for NoiGAN. Please find the result in Table 3 in our latest version of the paper. The results show that both NoiGAN-TransE and NoiGAN-RotatE consistently and significantly outperform all the baseline methods in terms of robustness.
>
> (2) Thanks to the reviewer for pointing out the problem of data leakage in FB15K and WN18. We have conducted the experiments on the FB15K-237 and WN18RR instead. Please find the result in Table 3 in our latest version of the paper.
>
> (3) Thanks to the reviewers for pointing out more related works. According to [1] pointed out by the reviewer, although the authors include real-world noise in the Biological Knowledge Graph, it still introduces random noise to FB15k-237, which is the same as what we do. The major reason is that the real-world noise is unavailable for the benchmark Knowledge Graph dataset, including FB15k-237, YAGO3-10 and WN18RR. Some of the other related works also use the same strategy to introduce random noise, such as [7], [8], [9].
>
> (4) We apologize for the unclear claim. We agree that well trained KGE models are widely used for denoising when constructing a knowledge graph, such as [2] mentioned by the reviewer. However, in order to achieve a reliable well trained KGE model, the training data has to be clean. The major reason is that current KGE models highly rely on high-quality training data and thus are lack of robustness to noise [7]. Given the fact that a real knowledge graph will inevitably include many kinds of errors, such as ambiguous, conflicting and erroneous and redundant information, it is difficult for us to find an ideal clean dataset to train KGE models. To address this problem, in this paper, we proposed a novel technique to enable current embedding models to cope with noisy data.
>
> [1]  “Interpretable Graph Convolutional Neural Networks for Inference on Noisy Knowledge Graphs.” Workshop at NeurIPS’18.
> [2]  “Knowledge Vault: A Web-Scale Approach to Probabilistic Knowledge Fusion.” KDD’14.
> [3]  “Kbgan: Adversarial learning for knowledge graph embeddings.” NAACL’18.
> [4]  “RotatE: Knowledge Graph Embedding by Relational Rotation in Complex Space.” ICLR'19.
> [5]  “Embedding Entities and Relations for Learning and Inference in Knowledge Bases”  ICLR'15.
> [6]  “Learning Attention-based Embeddings for Relation Prediction in Knowledge Graphs”, ACL’19.
> [7]  “Sparsity and noise:  Where knowledge graph embeddings fall short.” ENMLP’17.
> [8]  “Does william shakespeare really write hamlet?  knowledge representation learning with confidence.” AAAI’18.
> [9]  “Confidence-aware negative sampling method for noisy knowledge graph embedding.” ICBK’18.

---

### Official Review · AnonReviewer4 · 2019-11-06
**Official Blind Review #4**

**Rating:** 3

**Review:**

This paper presented a jointly learning framework based on GAN for tackling both knowledge graph completion and noise detection simultaneously. Existing works only deal with each of task independently and did not investigate the benefits of coping with both tasks together. The paper is well motivated. In order to achieve them, the paper presented a GAN framework in order to train a noising KG embedding as well the generator and discriminator. The key connections between two parts are through the confidence of a noise triple and generation of the negative sample triples. The whole framework looks quite interesting and promising. The experimental results are provided to validate the effectiveness of the proposed model.

There are two key concerns about this paper:

1) It is well known that both GAN and RL are hard to train, not to mention combining them together to joint train in order to deal with data indifferenceability issue of discrete triple generation. Are the results easy to reproduce?

2) Choosing 10% triples as positive training examples seems very ad-hoc. Have you studied the sensitivity of the number of percentage of triples as positive training examples on the system performance?

3) I don't know too much about the methods from knowledge graph noise detection so maybe one baseline - CKRL is enough for representing state-of-the-arts. However, for knowledge graph completion task, TransE is most simple baseline and they are rich state-of-the-art methods in this line such as [1]. It is not convincing to show the advantages of the proposed NoiGAN without such comparisons.

[1]  “RotatE: Knowledge Graph Embedding by Relational Rotation in Complex Space.” ICLR'19.




**Experience Assessment:**

I have read many papers in this area.

**Review Assessment: Checking Correctness Of Derivations And Theory:**

I assessed the sensibility of the derivations and theory.

**Review Assessment: Checking Correctness Of Experiments:**

I carefully checked the experiments.

**Review Assessment: Thoroughness In Paper Reading:**

I read the paper at least twice and used my best judgement in assessing the paper.

---

> ### Author Response · Authors · 2019-11-15
> **Responds to Review #4**
>
> We thank the reviewer for the constructive reviews. We addressed the questions and concerns of the reviewer accordingly in the following.
>
> 1 Using policy gradient to generate discrete data with GAN is first proposed by [7] and [2] and shows great performance in information retrieval in [2]. Afterward, this strategy has been widely adopted to learn graph representation, e.g., [1] and [3]. Following [1], [2], [3], [7],  we adopt the same strategy in our work. We agree that GAN is unstable and hard to train. Fortunately, in our work, it doesn’t cause much trouble. To show that our model is stable and our results are easy to reproduce, we train NoiGAN-RotatE (soft) with the same parameters for 3 times on FB15K-237 with 70% noise and report the results on test dataset as follows:
>
> MRR  HITS@1  HITS@3  HITS@10
> 0.279   0.179       0.320       0.475
> 0.279   0.179       0.319       0.477
> 0.279   0.179       0.319       0.474
>
> We can observe that the results are almost the same, which shows the stability of our model.
>
> (2) To study the effect of the percentage of triples as positive training examples, we also run NoiGAN-RotatE (soft) with the percentage of triples as positive training examples as 10% 20% 40% on FB15K-237 with 70% noise, the result is as follows:
>
> Percentage of triples  MRR  HITS@1  HITS@3   HITS@10
> 10%                               0.279   0.179       0.320         0.475
> 20%                               0.278   0.179       0.318         0.473
> 40%                               0.279   0.180       0.318         0.475
>
> We can observe that the variation among the results is relatively small. It indicates that our NoiGAN is less sensitive to the percentage of triples as positive training examples.
>
> (3) Thanks to the reviewer for pointing out this issue. We have added more baseline methods, including (1) KGE models (e.g., DistMult [5] and RotatE [4]), (2) robust KGE models (e.g., attention based method [6]) and (3) KGE models with GAN (e.g., KBGAN [3]). In addition, to show that our NoiGAN can be easily generalized to various KGE models, RotatE is also added as score function for NoiGAN. Please find the results in Table 3 in our latest version of the paper. The results show that both NoiGAN-TransE and NoiGAN-RotatE consistently and significantly outperform all the baseline methods in terms of robustness.
>
> [1]  “GraphGAN: Graph Representation Learning with Generative Adversarial Nets.” AAAI'18.
> [2]  “Irgan: A minimax game for unifying generative and discriminative information retrieval models.” SIGIR'17.
> [3]  “Kbgan: Adversarial learning for knowledge graph embeddings.” NAACL’18.
> [4]  “RotatE: Knowledge Graph Embedding by Relational Rotation in Complex Space.” ICLR'19.
> [5]  “Embedding Entities and Relations for Learning and Inference in Knowledge Bases”  ICLR'15.
> [6]  “Learning Attention-based Embeddings for Relation Prediction in Knowledge Graphs”, ACL’19.
> [7]  “SeqGAN: Sequence Generative Adversarial Nets with Policy Gradient”, AAAI’17.

---

### Decision · Program_Chairs · 2019-12-19

**Decision:**

Reject

**Comment:**

This paper proposes a noise-aware knowledge graph embedding (NoiGAN) by combining KG completion and noise detection through the GANs framework. The reviewers find that the idea is interesting, but the comparison to SOTA is largely missing. The paper can be improved by addressing the reviewer comments.